# Minimally Invasive Versus Open Distal Gastrectomy for Locally Advanced Gastric Cancer: Trial Sequential Analysis of Randomized Trials

**DOI:** 10.3390/cancers16234098

**Published:** 2024-12-06

**Authors:** Alberto Aiolfi, Matteo Calì, Francesco Cammarata, Federica Grasso, Gianluca Bonitta, Antonio Biondi, Luigi Bonavina, Davide Bona

**Affiliations:** 1IRCCS Ospedale Galeazzi–Sant’Ambrogio, Division of General Surgery, Department of Biomedical Science for Health, University of Milan, Via C. Belgioioso, 173, 20157 Milan, Italy; matteocali94@gmail.com (M.C.); fra.cammarata@hotmail.it (F.C.); federigrasso@gmail.com (F.G.); bbonit@icloud.com (G.B.); davide.bona@unimi.it (D.B.); 2G. Rodolico Hospital, Surgical Division, Department of General Surgery and Medical Surgical Specialties, University of Catania, 95131 Catania, Italy; abiondi@unict.it; 3IRCCS Policlinico San Donato, Division of General and Foregut Surgery, Department of Biomedical Sciences for Health, University of Milan, 20097 Milan, Italy; luigi.bonavina@unimi.it

**Keywords:** open distal gastrectomy, minimally invasive distal gastrectomy, locally advanced gastric cancer, long-term survival, trial sequential analysis

## Abstract

Minimally invasive distal gastrectomy (MIDG) has demonstrated benefits in short-term outcomes over open distal gastrectomy (ODG) for patients with locally advanced gastric cancer (LAGC), though its long-term survival impact is still uncertain. This systematic review and trial sequential analysis (TSA) examined randomized controlled trials (RCTs) reporting the long-term survival for MIDG vs. ODG. Five RCTs involving 2835 patients were analyzed, with 50.1% undergoing MIDG. The results indicated similar 5-year overall survival (OS) and disease-free survival (DFS) rates for both procedures. However, the TSA indicated that the accumulated evidence was insufficient to draw definitive conclusions about long-term survival outcomes. Thus, while MIDG and ODG appear comparable in 5-year OS and DFS for LAGC patients, further research is needed to corroborate our findings.

## 1. Introduction

Gastric cancer (GC) ranks the fifth highest cause of cancer-related fatalities globally [1,2]. Management strategies for GC have evolved in recent years, incorporating personalized multimodal therapies. However, surgery continues to be the cornerstone of treatment. For cancers diagnosed in the distal stomach, distal gastrectomy with D1+/D2 lymphadenectomy is considered the standard approach [3,4,5]. Historically, open distal gastrectomy (ODG) has been viewed as the optimal surgical method for both early gastric cancer (EGC) and locally advanced gastric cancer (LAGC) [6]. The advantages of minimally invasive distal gastrectomy (MIDG), which include reduced surgical trauma and improved precision in dissection, have contributed to its increasing acceptance among the surgical community. Numerous studies and randomized controlled trials (RCTs) have reported enhanced short-term and functional outcomes associated with MIDG in LAGC scenarios [7,8,9,10,11,12,13,14].

However, the effect of minimally invasive surgery on long-term survival in the settings of LAGC is still a topic of discussion [11,12,13,14,15,16,17,18,19,20,21,22,23,24]. Some trials have indicated that survival rates for MIDG are comparable or non-inferior to ODG. Additionally, recent meta-analyses have shown no significant differences in 5-year overall survival (OS) and disease-free survival (DFS) [25,26,27,28]. The objective of this updated systematic review was to evaluate the comparison between MIDG and ODG for LAGC through randomized controlled trials (RCTs) and to conduct a trial sequential analysis (TSA) to determine whether the necessary information size has been met, providing definite evidence, or if further studies are necessary to bolster actual knowledge.

## 2. Materials and Methods

A systematic review adhering to the preferred reporting items for systematic reviews and meta-analyses (PRISMA 2020) guideline [29] was reported without the need for ethical approval. PubMed, Scopus, MEDLINE, Web of Science, ClinicalTrials.gov, the Cochrane Central Library, and Google Scholar were utilized for the search process. The initial search took place in November 2023, was repeated in March 2024, and updated on 1 July 2024. Medical subject headings (MeSH terms) such as “gastric cancer”, “gastric neoplasm”, “gastric carcinoma”, “gastrectomy”, “distal gastrectomy”, “survival”, “overall survival”, “disease-free survival”, and “cancer specific survival” were combined with “AND” or “OR” (Appendix B). All titles were reviewed, relevant abstracts were extracted, and the reference lists of identified articles were independently assessed by three authors (MC, AA, and FC). The systematic review was registered with the PROSPERO database (CRD42024582264).

### 2.1. Eligibility Criteria

Inclusion criteria were as follows: (a) RCTs comparing ODG vs. MIDG for the treatment of LAGC; (b) RCTs reporting long-term OS, and DFS hazard ratios (HRs) or Kaplan–Meier curves; (c) in case of study overlap published by the same institution, study group, or dataset, we included the larger sample size or the most recent study. Exclusion criteria were as follows: (a) observational studies; (b) studies without a comparative analysis of ODG vs. MIDG; (c) studies principally reporting data for EGC or including >35% of EGC in the definitive pathological assessment; (d) studies that did not report any of the predefined primary outcomes; (e) studies not written in English.

### 2.2. Data Extraction

The collected data encompassed details about authors, publication year, country, trial acronym, patient count, gender, age, body mass index (BMI), American Society of Anesthesiologists (ASA) physical status, tumor characteristics, tumor location, surgical approach, follow-up duration, overall survival (OS), and disease-free survival (DFS). Three authors (MC, AA, FC) independently compiled all the data, which were reconciled at the end of the evaluation. A fourth author (DB) examined the database to resolve any discrepancies and ensure its accuracy.

### 2.3. Outcomes of Interest and Definitions

The primary outcomes included 5-year OS and DFS. OS was defined as the duration from surgery to the last known follow-up and death. DFS was defined as the period from surgical resection to the occurrence of local or systemic recurrence. Patient survival data were derived from HRs or Kaplan–Meier curves. Additional data were sourced from pertinent articles and related Appendix A. GC was defined as any primary histopathologically confirmed neoplasm situated in the antrum of distal gastric body. LAGC was defined as nonmetastatic cancer with wall invasion beyond the submucosa. Distal gastrectomy was defined as the anatomic surgical removal of the distal stomach with associated D1+/D2 lymphadenectomy. MIDG was defined as totally minimally invasive distal gastrectomy with intracorporeal anastomosis completion and alimentary tract reconstruction or laparoscopic-assisted distal gastrectomy with extracorporeal anastomosis and alimentary tract completion by mini laparotomy (<10 cm).

### 2.4. Quality Assessment and Assessment of Certainty of Evidence

The Cochrane risk-of-bias tool was used to assess the methodological quality of the identified RCTs [30]. Two authors independently assessed the methodological quality of the selected trials according to the following: (1) method of randomization, (2) allocation concealment, (3) baseline comparability of study groups, and (4) blinding and completeness of follow-up. Trials were then graded as having low (green circle), high (red circle), or unclear (yellow circle) risk of bias. We created GRADE evidence profiles to assess the certainty of each comparison and outcome using GRADEpro (https://www.gradepro.org; accessed on 15 September 2024). The certainty of the evidence is assessed based on factors such as risk of bias across studies, incoherence, indirectness, imprecision, publication bias, and other relevant parameters [31,32].

### 2.5. Statistical Analysis

The results of the systematic review were qualitatively summarized using a frequentist study-level random effects meta-analysis to calculate the pooled hazard ratio (HR). The analysis employed the inverse-variance method along with the DerSimonian–Laird estimator to assess the variance in the true effect size (τ^2^) [33]. Heterogeneity among studies was evaluated using the I^2^ index and Cochran’s Q test, where statistical heterogeneity was classified as low, moderate, or high based on I^2^ values of 25%, 50%, and 75%, respectively, and deemed significant when *p* < 0.10 [34,35,36,37]. A Wald-type 95% confidence interval (CI) was calculated for pooled measurements, while the 95% CI for the I^2^ index was computed following Higgins and Thompson [38]. The prediction interval for the treatment effect of a new study was derived according to Borestein [35]. Given the varying sample sizes across studies, a sensitivity analysis was conducted by excluding one study at a time and re-running the analysis to confirm the robustness of the overall results. A two-sided *p*-value of less than 0.05 was considered statistically significant. All analyses and figures were generated using R software, version 3.2.2 [39].

Additionally, a trial sequential analysis (TSA) was performed to evaluate the potential for type I error and to compute the information size [40,41]. The Lan–DeMets approach was used to establish monitoring boundaries and to set adjusted thresholds for statistical significance. The information size was calculated with α = 0.05 and β = 0.2, assuming a 25% risk ratio reduction (RRR) and a follow-up loss of 1% [42]. The z-curve was developed based on consecutive z-values calculated using two-sided significant testing. Monitoring boundaries were constructed using conventional testing and applying the O’Brien–Fleming α-spending function. The total number of observed patients in the cumulative meta-analysis was defined as the accrued information size (AIS), and the TSA was conducted using Stata 14 software [43].

## 3. Results

### 3.1. Selection Process

The flow chart illustrating the selection process is depicted in Figure 1.

Our preliminary search revealed a total of 53,033 publications. Subsequently, after removing duplicates, 4322 titles and abstracts were assessed. Further screening identified six studies that fulfilled the inclusion and exclusion criteria [12,13,14,16,21,24]. The studies conducted by Huang et al. and Yu et al. were consolidated into a single trial (CLASS-01) [12,16], as the former provided overall survival (OS) data while the latter reported disease-free survival (DFS) data for the same cohort of patients.

### 3.2. Quality Assessment

The comprehensive quality assessment according to the Cochrane risk of bias tool is illustrated in Appendix A [44]. All included randomized controlled trials (RCTs) exhibited issues relating to blinding, given the inherent challenges of implementing blinding in surgical RCTs, particularly in studies comparing interventions that result in different types of surgical incisions (e.g., minimally invasive versus open surgery) (Appendix A). The trial design was clearly defined in five trials, all of which reported a non-inferiority methodology. The randomization methodology was described in all trials, while four RCTs detailed the proficiency of the operating surgeons, including the number of previous ODGs or MIDs performed. Power analysis details were provided in four trials (one underpowered), whereas one study did not report any power analysis data. The systematic review and quantitative analysis were based on intention-to-treat (ITT) and per-protocol (PP) analyses for three trials [12,14,16] and ITT analysis for two trials [13,21], while one study did not specify the type of analysis utilized [24].

### 3.3. Systematic Review

Five trials from a total of six studies were included in the quantitative analysis, encompassing a combined total of 2835 patients. The demographic data are summarized in Table 1 [12,13,14,16,21,24]. Among the patients, 1421 (50.1%) underwent MIDG. The age of the patient population varied from 48 to 80 years, with a predominance of males (1768; 62.4%). Preoperative BMI varied from 17.3 to 27.3 kg/m^2^. Adenocarcinoma was diagnosed in 28.6% of patients. Pathologic tumor stage information was provided in all studies, revealing that 27.2% of tumors were stage I, 36.4% were stage II, 35.3% were stage III, and 1% were stage IV. Notably, none of the patients received neoadjuvant treatments; however, adjuvant treatment was reported in four studies, involving 1273 patients, and was performed according to different protocols/regimens.

### 3.4. Meta-Analysis and TSA

Four trials (2639 patients) reported 5-year OS data with comparable HR for MIDG vs. ODG (HR = 0.86; 95% CI 0.70–1.04) [12,13,14,24] (Figure 2A). The prediction lower and upper limits are 0.55 and 1.32, respectively. The heterogeneity is zero (I^2^ = 0.0%; 95% CI 0–85%; *p* = 0.78) and τ^2^ = 0.0. Visual inspection of the Funnel plot does not show evidence of publication bias. The sensitivity analysis shows the robustness of these results in terms of point estimation, heterogeneity, and confidence intervals. The TSA, assuming an anticipated intervention effect of 25% RRR, α = 0.05, and power (1 − β) = 0.96, shows a cumulative z-curve not crossing the monitoring boundaries line (Z = 1.96) (Figure 2B), thus suggesting a false negative 5-year OS HR because the meta-analysis included fewer patients than the required information size.

Five trials (2835 patients) reported 5-year DFS data with comparable HR for MIDG vs. ODG (HR = 1.03; 95% CI 0.87–1.23) [13,14,16,21,24] (Figure 3A). The prediction lower and upper limits are 0.78 and 1.38, respectively. The heterogeneity is zero (I^2^ = 0.0%; 95% CI 0–79%; *p* = 0.94) and τ^2^ = 0.0. Visual inspection of the Funnel plot does not show evidence of publication bias. The sensitivity analysis shows the robustness of these results in terms of point estimation, heterogeneity, and confidence intervals. The TSA, assuming an anticipated intervention effect, of 25% RRR, α = 0.05, and power (1 − β) = 0.97, shows a cumulative z-curve not crossing the monitoring boundaries line (Z = 1.96) (Figure 3B) thus suggesting a false negative 5-year DFS HR because the meta-analysis included fewer patients than the required information size.

### 3.5. Certainty of Evidence—GRADE

Full evaluation of the certainty of evidence and considerations for grading are detailed in the Appendix A. High certainty of evidence was found for 5-year OS and DFS.

## 4. Discussion

This meta-analysis indicates that MIDG with D1+/D2 lymphadenectomy seems comparable to ODG in term of 5-year OS and DFS when considering patients with LAGC. The TSA demonstrate that these data should be interpreted prudently since the global information size was not reached for both OS and DFS and this may prefigure a false negative meta-analytical result.

GC ranks as the sixth most common cancer and is the third leading cause of cancer-related deaths globally [1,2]. For patients with resectable GC, a multimodal treatment approach that integrates surgery and systemic therapy appears to improve survival rates. Surgical resection combined with lymphadenectomy remains the primary curative strategy for GC [3,4]. While ODG has traditionally been the standard treatment, MIDG has gained quick acceptance since its introduction in the early 1990s [6]. The advantages of MIDG, which minimizes surgical trauma, have contributed to its international widespread use. A recent nationwide survey in Korea has highlighted this shift, indicating a marked decrease in open surgeries and a corresponding rise in minimally invasive approaches [45]. Previous trials and meta-analyses have validated the safety and feasibility of minimally invasive techniques for EGC, showing improved short-term outcomes regarding surgical complications, recovery duration, and overall patient quality of life [7,8,9,10,15,22,46,47,48]. Similarly, recent studies have emphasized notable differences in short-term outcomes between ODG and MIDG in the setting of LAGC. These studies suggest that minimally invasive methods generally lead to fewer complications within 30 days post-surgery. Additionally, these techniques are linked to better postoperative pain management, enhanced patient comfort, and earlier mobilization [9,10]. Besides pain relief, patients may also experience a potentially reduced risk of postoperative complications, including infections, bleeding, and anastomotic leaks. A few analyses comparing the two approaches have noted that while traditional open surgery is still common, MIDG provides comparable oncological outcomes without sacrificing treatment effectiveness [7,8,9,10,15,46,47,48]. Another important finding is that patients who undergo MIDG often return to normal dietary habits more rapidly than those recovering from ODG, which not only boosts patient satisfaction but also accelerates overall recovery [9].

However, the effectiveness of minimally invasive treatments for LAGC has been debated due to concerns about laparoscopic manipulation with a potential increase in peritoneal cancer spread risk and early lymph node recurrence. The 5-year OS rates for LAGC vary by pathologic stage [14,16,49]. For stage IIB, the rate is approximately 50–60%, while stage IIIA shows a rate of about 30–40%. Patients with stage IIIB have a survival rate of around 15–25%, and those at stage IIIC may experience rates as low as 10–15%. Preoperative factors like age, comorbidities, and treatment response can affect long-term OS [50,51]. Additionally, intraoperative and postoperative complications, such as bleeding that necessitates transfusion [52], anastomotic/duodenal leak, and pulmonary complications, may also influence OS and DFS [53,54,55,56,57]. Notably, the impact of the surgical approach on patient survival remains a topic of significant debate, with some observational studies indicating that minimally invasive techniques may lead to better outcomes due to their presumed lower incidence of operation-related temporary immunodeficiency [58,59]. Recent trials and meta-analyses have demonstrated that MIDG yields non-inferior short-term outcomes, margin positivity rates, number of lymph nodes retrieved, and recovery times compared to ODG. Research from China, Korea, and Japan has provided 3-year and 5-year survival data indicating that MIDG achieves comparable OS and DFS rates to ODG [12,13,14,23]. Recently published meta-analyses also report similar long-term survival statistics, with Lei et al. showing equivalent 5-year OS (HR = 0.97; *p* = 0.73) and 5-year DFS (HR = 1.08; *p* = 0.64) for both minimally invasive and open gastrectomy [27]. Similarly, Davey et al. found no significant difference in disease recurrence (OR = 1.09, 95% CrI 0.76–1.49) and survival (OR = 1.02, 95% CrI 0.76–1.52) when comparing open and minimally invasive gastrectomy techniques [25].

Our analysis suggests that MIDG and ODG offer equivalent 5-year OS rates. At 5-year follow-up, the OS HR was estimated at 0.86 (95% CI 0.70–1.04). These results align with the JLSSG090118 trial [14], which showed non-inferior 5-year OS for MIDG compared to ODG in both intention-to-treat (HR = 0.83; *p* = 0.34) and per-protocol analyses (HR = 0.75; *p* = 0.19). The KLASS-02 trial, which analyzed data from 974 patients, also reported comparable 5-year OS rates (88.9% for MIDG vs. 88.7% for ODG) [13]. Similarly, the CLASS-01 trial from China indicated comparable 5-year OS estimates for D2 MIDG and ODG (72.6% vs. 76.3%) [12]. Similarly, a 2020 multicenter study from the U.S. and China using propensity score matching involving 889 patients found no significant difference in survival rates (54% for minimally invasive vs. 50.4% for open gastrectomy; *p* = 0.205) [60]. In our quantitative synthesis, the 5-year DFS HR estimation was similar for MIDG and ODG (HR = 1.03; 95% CI 0.87–1.23). This is consistent with the JLSSG090118 trial, which found comparable 5-year DFS rates for MIDG versus ODG in both ITT (75.7% vs. 73.9%) and PP analyses (81.4% vs. 78.4%) [14]. The KLASS-02 trial confirmed similar 5-year DFS for both techniques (81.1% for MIDG vs. 79.5% for ODG) [13]. These findings suggest that the surgical technique may not significantly influence survival, which is more likely determined by factors such as patient comorbidities, nutritional status at diagnosis, tumor stage and grade, histological variants, postoperative complications, lymphatic invasion, genetic susceptibility, molecular expression (including HER2 and PD-L1), mismatch repair, and microsatellite instability status, along with responsiveness to perioperative treatments [61]. Further, it has been proposed that the performance of surgeons with differing levels of training and experience might impact long-term survival following gastrectomy [62]. Specifically, Asplund et al. conducted a nationwide population-based cohort study in Sweden, involving 261 surgeons and 1636 patients, which concluded that surgeons who performed more than 20 gastric resections for adenocarcinoma demonstrated improved long-term survival rates [63]. Similarly, a study from South Korea indicated that surgeons who had conducted more than 100 resections for cancer achieved enhanced 5-year survival rates [64]. Finally, centralizing procedures in high-volume referral centers may improve 5-year survival and reduce recurrence rates [65,66,67,68]. Therefore, despite the zero heterogeneity (I^2^ = 0.0%), also confirmed by the sensitivity analysis, all these potential confounders should be considered as potential bias in our meta-analysis.

Given the low heterogeneity for both outcomes (I^2^ = 0%), we performed a TSA [40,42]. This is a complex statistical approach employed to evaluate evidence from clinical trials. It aims to determine whether the aggregated evidence from various studies is adequate for making reliable conclusions regarding treatment effects while managing both Type I and Type II errors. By integrating various interim analyses, TSA facilitates the calculation of the number of patients required to sufficiently power the analysis and reduce the risk of incorrect conclusions [69]. In our analyses, concentrated on OS and DFS, the cumulative z-curve failed to cross the boundary Z line, meaning that the TSA did not achieve the necessary information size to provide conclusive evidence. Consequently, both outcomes may prefigure false-negative findings. Therefore, additional trials focusing on 5-year OS and DFS in patients with LAGC are essential for gathering further evidence that could either support or challenge our meta-analytical results.

Several essential issues should be noted regarding the included trials. First, the 2018 Korean COACT1001 is a Phase II trial rather than a true Phase III trial, which leads to underpowered long-term survival results [21]. Additionally, while the KLASS-02 study is classified as Phase III, multiple publications have emerged, highlighting slight variations in the statistical methods used across these papers [13]. In contrast, the 2023 Japanese JLSSG0901 trial is a fairly rigorous prospective non-inferiority phase III trial conducted by certified surgeons (ESSQS certification) [14]. In the CLASS-01 trial, there was higher incidence of anastomotic leakage in MIDG patients, suggesting a potential selection bias [12,16]. Further, operative times and blood loss were comparable in both CLASS-01 and KLASS-02 [12,13,16]. In contrast, JLSSG0901 reported significantly less blood loss with MIDG, although the operative time was considerably longer, indicating that minimally invasive surgical approaches may have varied among trials and operating surgeons [14]. Unfortunately, these variations cannot be quantified or effectively stratified in the quantitative analysis.

To our knowledge, this is the first TSA evaluating the impacts of MIDG compared to ODG. To enhance our evidence, we specifically included RCTs that concentrated exclusively on distal gastric resections, thereby reducing the heterogeneity (I^2^ = 0.0%) that could result from studies involving total gastrectomy and EGC. While considering our findings, it is important to note some limitations. First, all studies were sourced from Eastern tertiary centers, which might limit the generalizability of the results to Western populations (LOGICA trial) [70] (please see the new added reference No.70). Second, although every study was an RCT, only a handful provided adequate detail regarding trial design, randomization procedures, blinding, power analysis, and third-party quality control of surgical procedures. Third, no patients in the studies received neoadjuvant therapy, and the reporting of adjuvant treatments varied across the trials, with inconsistent protocols and regimens. Fourth, even with thorough preoperative assessments, about 25% of patients initially classified as LAGC were down-staged to EGC upon pathological examination. Lastly, various factors such as differences in multidisciplinary perioperative care teams, operator skill levels and hospital volumes, complications, genetic expression, and molecular profiles should be considered due to their non-negligible influence on long-term patient survival.

## 5. Conclusions

MIDG and ODG appear to exhibit equivalent 5-year OS and DFS in LAGC patients. However, the cumulative evidence from the TSA indicates that the actual information size is inadequate for drawing definitive conclusions, thereby configuring a potential false negative result. Future trials should prioritize long-term OS and DFS assessments in patients with LAGC.

## Figures and Tables

**Figure 1 cancers-16-04098-f001:**
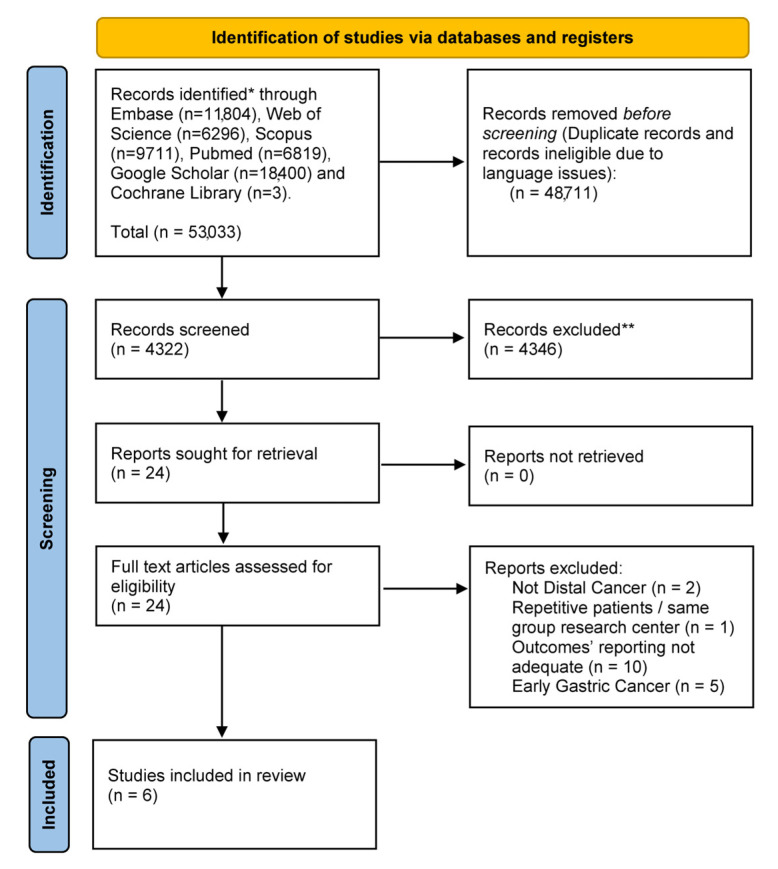
The preferred reporting items for systematic reviews and meta-Analyses (PRISMA) diagram.

**Figure 2 cancers-16-04098-f002:**
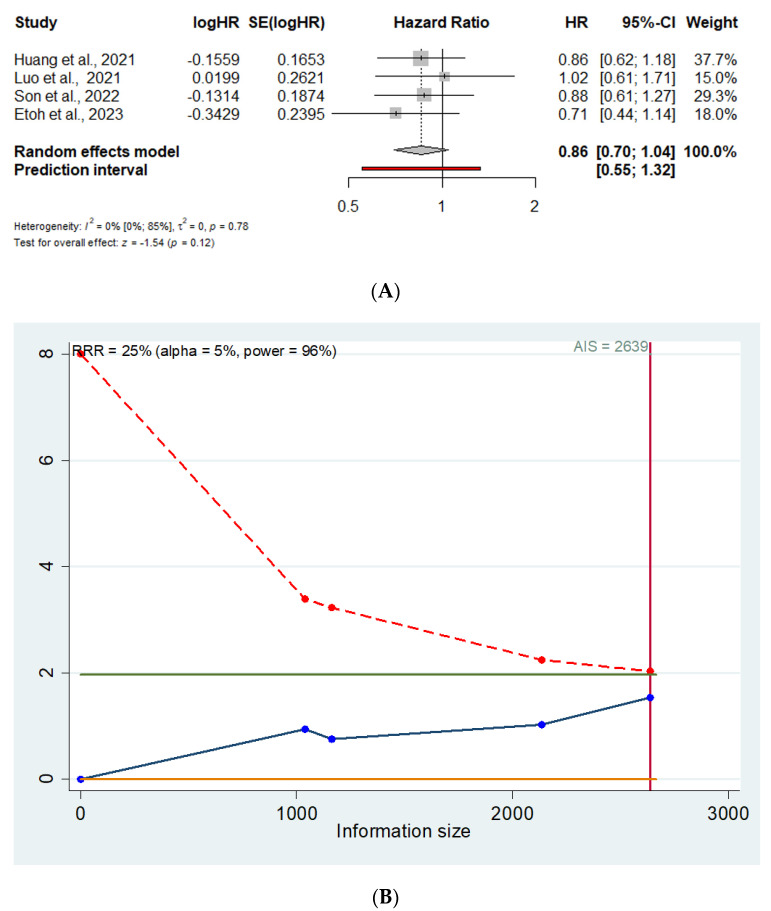
(**A**). Forrest plot for 5-year OS. HR: hazard ratio; 95% CI: confidence interval. (**B**). Trial sequential analysis for 5-year OS. X axis: Information size (number of patients); Y axis: Cumulative Z score; AIS: Accrued information size; Green line: Z=1.96; Red line: Monitoring boundary line; Blue line: Trial sequential analysis line [12,13,14,24].

**Figure 3 cancers-16-04098-f003:**
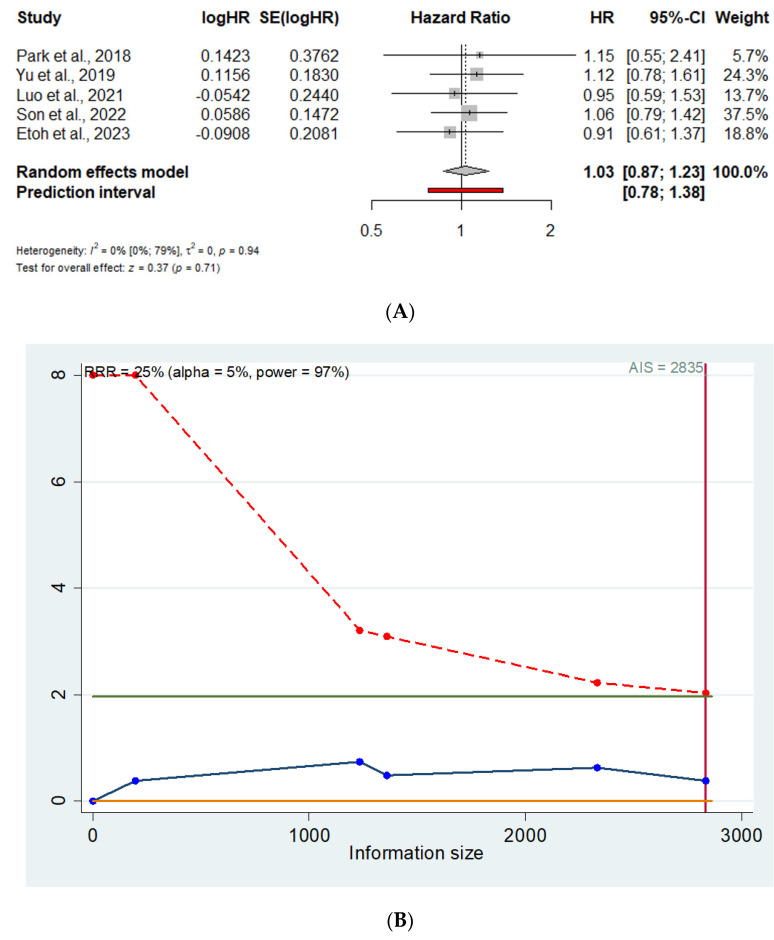
(**A**). Forrest plot for 5-year DFS. HR: hazard ratio; 95% CI: confidence interval. (**B**). Trial sequential analysis for 5-year DFS. X axis: Information size (number of patients); Y axis: Cumulative Z score; AIS: Accrued information size; Green line: Z=1.96; Red line: Monitoring boundary line; Blue line: Trial sequential analysis line [13,14,16,21,24].

**Table 1 cancers-16-04098-t001:** Demographic and clinical characteristics of patients undergoing ODG and MIDG; not reported (NR); AJCC American Joint Committee on Cancer; adenocarcinoma (A); signet ring cell carcinoma (SRC); intestinal histological type (I); diffuse histological type (D); differentiate (DF); undifferentiated (UDF); others (O). Data are reported as numbers, mean ± standard deviation, median (range).

Author, Year, Country	Trial Acronym	Study Period	Surgical Procedure	No. Patients	Age (yrs)	Gender (M/F)	BMI (kg/m^2^)	Staging System	Stage Ia	Stage Ib	Stage II	Stage III	Stage IV	Tumor Histology	Adjuvant
Park et al., 2018, Korea [21]	COACT1001	2010–2011	ODG	96	60.1 ± 8.2	65/31	23.3 ± 3.1	AJCC 7th	22	14	33	23	4	196 A	78
			MIDG	100	58.6 ± 8.9	69/31	23.7 ± 3.0		27	15	29	28	1		77
Luo et al., 2021, China [24]	nr	2008–2012	ODG	62	63.9 ± 15.4	43/19	21.5 ± 4.2	nr	0	5	14	43	0	58A, 4SRC	57
			MIDG	62	64 ± 15.2	42/20	21.5 ± 4.2		0	5	13	44	0	55A, 7SRC	55
Yu et al., 2019, China [16] and Huang et al., 2021, China [12]	CLASS-01	2012–2017	ODG	520	55.8	346/174	22.7	Japanese and AJCC 7th	88		247	185	0	99SRC, 421O	217
			MIDG	519	56.5	380/139	22.7		64		248	207	0	79SRC, 440O	192
Son et al., 2022, Korea [13]	KLASS-02	2011–2015	ODG	482	59.4	335/147	23.7	AJCC 8th	165		167	150	0	187DF, 278UDF, 17O	299
			MIDG	492	59.8	151/141	23.5		178		148	166	0	197DF, 286UDF, 9O	298
Etoh et al., 2023 Japan [14]	JLSSG0901	2009–2016	ODG	254	67 (33–80)	168/86	22.5	AJCC 7th	45	50	69	78	12	502A	nr
			MIDG	248	64 (34–80)	169/79	22.3		53	39	65	78	13		

## Data Availability

Data generated at a central, large-scale facility are available upon request from the corresponding author.

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
