# Peer review of "Minimally Invasive Versus Open Distal Gastrectomy for Locally Advanced Gastric Cancer: Trial Sequential Analysis of Randomized Trials"

_cancers, 2024, doi:10.3390/cancers16234098_

Round 1

Reviewer 1 Report

Comments and Suggestions for Authors

The authors performed a meta-analysis using trial sequential analysis to determine the effect of MIDG vs. ODG on long-term patients’ survival.

This study is an interesting attempt that focuses on an important point. This is an excellent paper.

However, please share the following points with me (the reviewer) and add them to the discussion.

1. Of the five trials listed here, COACT1001 is a small Phase II trial, not strictly a Phase III trial.

2. Although the KLASS-02 study is a phase III study, many papers have been published, and it has been pointed out that the statistical methods differ slightly between these papers. In CLASS-01, LDG had more anastomotic leakage, and it was pointed out that the cases included in the final analysis may have been selected. In addition, the operative time and blood loss in CLASS-01 and KLASS-02 were similar. Still, the blood loss in LDG in JLSSG0901 was significantly less, and the operative time was much longer, so the surgery content may have differed considerably among the three studies. 

Therefore, the LDG was not necessarily uniform among these five RCTs. Is it possible that there were fluctuations in the meta-analysis, leading to the result of missing cases in TSA?

3. Please note that JLSSG0901 was a fairly rigorously conducted prospective non-inferiority phase III trial. Based on the results of the JLSSG0901, it may be safe to conclude that LDG and ODG are equivalent.

Author Response

The discussion section has been revised accordingly. As you mentioned, there may be inherent inter-operator variabilities that contribute to differences in the MIDG techniques used in the included trials. Unfortunately, these variations cannot be quantified or stratified effectively in the TSA. A dedicated paragraph addressing all these issues has been added. Thank you for highlighting these concerns and for the possibility to improve our manuscript.

Reviewer 2 Report

Comments and Suggestions for Authors

The article is supported by a robust statistical anlysis; nevertheless, the aspected results are still inconclusive. This is due, in my opinion, to the extreme eterogeneity of the analysed trials, with different patients, different follow up lenght and surgical techniques. Furthermore, only 28.6% of the MIDG presented with a diagnosis of adenocarcinoma, and 25% of them were EGC; none of the patients had neoadjuvant therapy. This condition deeply affects the significativity of the results. I think those limits should be emphasized in the article.

I don't believe the reference n. 68 is relevant for the article

Comments on the Quality of English Language

There are some spelling errors that must be emended

Author Response

We completely concur with your points. All these limitations have been addressed in the dedicated section of the manuscript, which has been updated accordingly. Reference 68 has been relocated to the statistical analysis section, and the grammar has been corrected as well. Thank you

Reviewer 3 Report

Comments and Suggestions for Authors

This paper addresses the question whether minimally invasive distal gastrectomy for locally advanced gastric cancer is comparable to open distal gastrectomy in terms of overall and disease-free long-term survival.

This review is beautifully written and of timely importance despite not reaching a definitive conclusion. It provides a concise reporting and is methodologically sound.

I only have a few questions and comments:

Minimally invasive includes both, robotic and laparoscopic procedures?

Records excluded 4346 of 4322?

Unfortunately, there were no patients with neoadjuvant treatment

Were recurrences local or rather systemic?

Author Response

R#3. This paper addresses the question whether minimally invasive distal gastrectomy for locally advanced gastric cancer is comparable to open distal gastrectomy in terms of overall and disease-free long-term survival. This review is beautifully written and of timely importance despite not reaching a definitive conclusion. It provides a concise reporting and is methodologically sound.

I only have a few questions and comments:

Minimally invasive includes both, robotic and laparoscopic procedures?

  1. A. In the included trials only laparoscopic procedures have been performed. At the moment there are no published trials reporting long-term survival data for robotic distal gastrectomy in the setting of LAGC.

Records excluded 4346 of 4322?

  1. A. We apologize for the mistake. This was a typo that has been amended accordingly.

Unfortunately, there were no patients with neoadjuvant treatment

  1. A. We totally agree. This is a common problem of all Eastern trials. This issue has been commented in the appropriate limitation section.

Were recurrences local or rather systemic?

  1. A. As defined in the methods section DFS survival was defined as the period from surgical resection to the occurrence of local or systemic recurrence. Thank you for highlighting all these concerns and for the possibility to improve our manuscript.